# Pathological Appearance of Focal Liver Reactions after Radiotherapy for Hepatocellular Carcinoma

**DOI:** 10.3390/diagnostics12051072

**Published:** 2022-04-25

**Authors:** Masahiro Okada, Kazushi Numata, Hiromi Nihonmatsu, Kengo Tomita, Atsuya Takeda, Kenichiro Tago, Tomoko Hyodo, Takahisa Eriguchi, Masayuki Nakano

**Affiliations:** 1Department of Radiology, Nihon University School of Medicine, Itabashi-ku, Tokyo 173-8610, Tokyo, Japan; tago.kenichiro@nihon-u.ac.jp; 2Gastroenterological Center, Yokohama City University Medical Center, 4-57 Urafune-cho, Minami-ku, Yokohama 232-0024, Kanagawa, Japan; kz-numa@urahp.yokohama-cu.ac.jp (K.N.); h.n.twopines@gmail.com (H.N.); 3Division of Gastroenterology and Hepatology, Department of Internal Medicine, National Defense Medical College, 3-2 Namiki, Tokorozawa-shi 359-8513, Saitama, Japan; kengo@boreas.dti.ne.jp; 4Radiation Oncology Center, Ofuna Chuo Hospital, 6-2-24, Ofuna, Kamakura 247-0056, Kanagawa, Japan; takeda@1994.jukuin.keio.ac.jp (A.T.); eriguchi@rad.med.keio.ac.jp (T.E.); 5Department of Radiology, Kindai University Faculty of Medicine, 377-2 Ohno-Higashi, Osaka-Sayama 589-8511, Osaka, Japan; ttst0801@gmail.com; 6Department of Pathology, Yokohama City University Medical Center, 4-57 Urafune-cho, Minami-ku, Yokohama 232-0024, Kanagawa, Japan; m-nakano@tcpl.co.jp

**Keywords:** stereotactic body radiotherapy, hypofractionated radiotherapy, focal liver reaction, hepatocellular carcinoma, pathology

## Abstract

We studied five pathological specimens from five patients at 1.5, 3.0, 4.0, 13.5, and 14.0 months after radiotherapy for HCC. Four needle biopsies were obtained to investigate liver parenchyma of focal liver reaction (FLR) around treated HCC, when patients had newly developed HCC or local recurrence appeared in the liver. Liver resection was performed in one case where insufficient radiotherapy effect for HCC was suspected. In all patients, FLR was recognized as a hypervascular area around the HCC on enhanced CT and enhanced Gd-EOB-DTPA (EOB-MRI). Liver specimens were analyzed to assess the pathological characteristics of FLR. FLR was recognized as prolonged liver enhancement in enhanced CT and EOB-MRI. From pathological understanding, sinusoidal dilatation with degeneration and desquamation was caused by direct endothelial cell injury following radiotherapy. Hepatocytes and endothelium fell off, and so the portal tract came close, and hepatic arteries increase simultaneously, resulting in FLR around HCC after radiotherapy. In conclusion, the prolapse of hepatocytes and sinusoidal endothelium induced neovascularization of hepatic arteries due to the repair mechanisms; in addition, these prolapse may shorten the distance between each portal region and the hepatic arteries flowing through the portal region become more prominent in FLR.

## 1. Introduction

Stereotactic body radiotherapy (SBRT) and hypofractionated radiotherapy (HFRT) are relatively new treatments for hepatocellular carcinoma (HCC), and Takeda et al. found excellent local control and overall survival after SBRT for HCC [1,2]. Herfarth et al. reported a change in the image of the surrounding liver parenchyma for the irradiated HCC, defined as focal liver reaction (FLR), on contrast-enhanced CT at 1.8 months after radiation therapy [3]. Shiozawa et al. [4] and Sanuki et al. [5] stated that, in cases of hepatic parenchyma surrounding the HCC, well-demarcated hyper-enhancement appeared in the vascular phase, followed by a hypo-echoic area in the post-vascular phase of contrast-enhanced ultrasound [4], which is referred to as FLR on CT [5]. In contrast, gadolinium-ethoxybenzyl-diethylenetriamine pentaacetic acid (Gd-EOB-DTPA; Primovist^®^; and Bayer Schering Pharma AG, Berlin, Germany)-enhanced MR imaging (EOB-MRI) represent the shape of well-demarcated focal hyperintensity in the arterial phase and hepatocyte hypofunction in FLR [6]. The volume of FLR decreased over time, as imaging changes in the surrounding liver of HCC began to appear at a median of 3 months after SBRT, peaked at a median of 6 months, and disappeared 9 months later [5]. Thus, FLR may be related to background liver function, but the underlying pathology is poorly understood, although it has been stated that increased arterial mechanism of FLR and injured endothelial cells occur in the irradiated liver [7,8]. Therefore, this study aimed to assess the pathological appearance of FLR in relation to radiotherapy toxicity in the context of HCC.

## 2. Materials and Methods

### 2.1. Patients

This retrospective study was approved by the institutional review board (B180900067). Written informed consent was obtained from all patients. All clinical investigations were conducted according to the principles of the Declaration of Helsinki. 

Five cases (Table 1) sharing the following FLR features on imaging are reported: (*a*) needle biopsy in four cases at 1.5, 3.0, 13.5, and 14.0 months after radiotherapy for HCC, and (*b*) resected liver specimen in one case at 4.0 months after radiotherapy for HCC.

### 2.2. Albumin-Bilirubin (ALBI) Scores

The albumin-bilirubin (ALBI) [9] and Child–Pugh scores were assessed to investigate liver function. The ALBI score was calculated as follows: score = (log_10_ total-bilirubin [mg/dL] × 17.1 × 0.66) + (albumin [g/dL] × 10 × −0.085) [9]. As previously reported, [10] patients were categorized as follows: grade 1 (score ≤ −2.60), grade 2a (−2.60 < score < −2.27), grade 2b (−2.27 ≤ score ≤ −1.39), and grade 3 (score > −1.39). 

### 2.3. Radiotherapy 

The patients were treated with SBRT or HFRT. When exceeding the constraint of the GI tract in SBRT planning (the maximum dose to a 3 mm isotropic expanded area of the GI tract was limited to <25 Gy in 5 fractions), patients were treated with HFRT with a total dose of 36–45 Gy in 12–15 fractions over 16–21 days. The other patients were treated with SBRT with a total dose of 35–54 Gy in 3–5 fractions over 3–7 days. Treatment was planned to enclose the planning target volume with a 60–80% isodose line of the maximal dose.

### 2.4. Imaging Methods

A dynamic contrast-enhanced CT (CECT) was performed using a 16 multi-detector CT (Aquilion-16; Canon Medical, Tochigi, Japan). The reconstruction section and the interval thickness were 5 mm. A nonionic contrast medium was intravenously injected using 100 mL of iopamidol (Iopamiron^®^ 300 or 370; Bayer Schering Pharma AG, Berlin, Germany) at 3 mL/s via a 20–22-gauge cannula. The scan timing of the arterial phase was determined using a bolus-tracking program (RealPrep^®^, Canon Medical, Tochigi, Japan), and the trigger point was set to 230 HU from the baseline attenuation of the abdominal aorta. The scanning times of the portal and equilibrium phases were set to 70 s and 180 s after the initiation of the injection.

EOB-MRI was performed using a 1.5 Tesla whole-body imager (Avant; Siemens Medical System, Erlangen, Germany), and Gd-EOB-DTPA was intravenously injected at 0.1 mL/kg and 1 mL/s through the antecubital vein. Next, 20 mL of sterile saline solution was delivered after Gd-EOB-DTPA injection. Arterial phase, portal phase, late phase, and hepatobiliary phase scanning were carried out at 25–30 s, 70–85 s, 180 s, and 20 min after contrast injection, respectively. 

### 2.5. Image Evaluation

The imaging findings were confirmed by a gastroenterologist and a radiologist with a consensus. Two observers evaluated the vascularity of the lesion and liver parenchyma in the arterial and portal phases on dynamic CECT and EOB-MRI, in addition to the hepatobiliary phase of EOB-MRI.

### 2.6. Pathology Materials and Pathological Analysis 

Pathological specimens were obtained from newly developed HCC or local recurrence, irradiated HCC, surrounding non-tumor areas (which is consistent with FLR images on CT and MRI), and non-irradiated non-tumor areas in each patient. Pathological analysis was performed in four cases of needle biopsy and one surgical operation case, a total of five cases. Specimens may only be obtained from local perspectives by needle biopsy, which is a technique that is precise and difficult to conduct. At first, we studied it from an operation specimen with the assumption that there was a difference in the influence of radiation therapy, dependent on distance from HCC, and that the strength of the radiation and the relations of the change in the liver may be clarified.

The liver tissue was observed by hematoxylin and eosin staining firstly, and then with a reticulin fiber net with a silver impregnation specimen. We examined the endothelium in cluster of differentiation (CD)31, CD34 and arteriole in αSmooth Muscle Actin (SMA) immuno-histochemical staining. The influence of SBRT was generally divided into three points: (1) closest, (2) mediated in vicinity of HCC, and (3) distal (no influence) at a distance from HCC.

### 2.7. Statistical Analysis

A paired *t*-test was performed to investigate the difference in ALBI scores between pre- and post-radiation therapy using commercially available software (IBM SPSS Statistics, Version-26.0, IBM Corp., Armonk, NY, USA). The statistical significance was set to *p* < 0.05.

## 3. Results 

### 3.1. Patient Characteristics 

Patient characteristics are represented in Table 1. Irradiated HCCs were well-differentiated HCCs in two cases, poorly differentiated HCCs in two cases, and pathologically unknown (imaging diagnosis based on AASLD guideline [11]) in one case. The median tumor size of HCC was 2.2 cm (range, 1.3–3.2 cm). Radiofrequency ablation (RFA) was performed for three cases, because three cases had local recurrence for irradiated HCCs. The liver biopsy was performed in the imaging FLR area, when RFA was performed. No clinical or laboratory findings were suggestive of portal hypertension. Case 5 had a transcatheter arterial chemo embolization (TACE), as a previous therapy. Case 4 had another HCC after radiotherapy, and the liver biopsy was done including FLR area.

In Case 3, percutaneous transhepatic portal vein embolization (PTPE) was performed for the right lobe of the liver 2 months prior to SBRT, because this patient had HCCs in segment 5 and segment 8 of the liver, which were scheduled to be resected together in the right lobe. However, the enlargement of the left lobe of the liver was not sufficient; thus, the strategy was changed to resection of an HCC in segment 5 and SBRT of an HCC in segment 8. The alfa-fetoprotein (AFP) and protein induced by vitamin K absence or antagonist-II (PIVKA-II) levels changed from 8.5 and 47% to 11.1 and 33%, respectively, after SBRT. On dynamic CECT, this patient had FLR around the HCC at 2.5 months after SBRT, and the FLR was consistent with the irradiated area. The HCC size did not change after SBRT, and the tumor stain of an HCC in segment 8 was residual at 2.5 and 4 months after SBRT. Therefore, right lobe resection was performed for HCCs in segments 5 and 8 after re-PTPE to the right lobe of the liver. Following liver resection, a poorly differentiated HCC was diagnosed as HCC in segment 8 on pathological assessment (Table 1).

### 3.2. Imaging Follow-Up

The median imaging follow-up period after radiation therapy was 33.6 months (range: 21–51 months). Follow-up every 3 months from 1 month after radiotherapy was continued for dynamic CECT, and occasionally, EOB-MRI was performed, as feasible. FLR was seen at 4 months after radiotherapy in all five cases, and FLR was seen at 10 months in two cases and at 13 months in one case. After 13 months, the interval between CT and MR examinations was found to be variable because imaging was performed for clinical demands. Even 4.0 months after SBRT, EOB-MRI showed FLR around irradiated HCC (Figure 1).

### 3.3. Albumin-Bilirubin (ALBI) Score and Grade

The liver function by ALBI score showed a decrease in four cases (Table 1), although one case showed no interval change in the ALBI score. The ALBI score was significantly (*p* = 0.040) worsened post-radiation therapy (−2.24) as compared to pre-radiation therapy (−2.44) by a paired *t*-test.

### 3.4. Pathological Findings and Estimated Theory of FLR Mechanisms

Most short terms after radiation were 1.5 months (Case 1). In the biopsy specimen of HCC, there is non-cancer tissue fragment included. In hematoxylin and eosin staining, the background liver after SBRT has a degenerated focus (Figure 2A). In the silver impregnation preparation, some hepatocytes showed fallout, and the reticulum frame structure was enlarged (Figure 2B *). The next short term after SBRT was 3 months (Case 2). Three-point biopsies (non-tumor, S8 SBRT area, and S5 early HCC suspected area) were performed. In the liver biopsy of the S8 SBRT area, irregular reticulin frame work and falling out of hepatocyte were observed in the silver impregnation stain. We confirmed whether it was the liver that received radiation with the liver tissue histology of silver impregnation stain pattern of the sinusoidal wall (Figure 3A). The background liver tissue fragment (Figure 3B right) may be included in biopsy specimens taken for the diagnosis of HCC in Case 2. At first, we compared the liver tissue with radiation effect and that without. We then noticed two sinusoidal changes: (1) endothelium disappearance (Figure 3C) and (2) the irregularity of the silver impregnation reticulum frame pattern. 

Silver impregnation staining showed atrophy and the disappearance of hepatocyte (Figure 4A). In the case of surgical material (Case 3, 4M), close to the HCC nodule, there was disappearance of liver parenchyma (hepatocytes and endothelium), so portal tracts came close (Figure 4A). The area was slightly beyond the HCC nodule. The radiation effect on the endothelium appeared rough, indicating degeneration (Figure 4B). Some parts of the endothelium disappeared, and the sinusoid was enlarged (Figure 4B). In the portal tract, the portal vein disappeared and the arteriole increased (Figure 4C).

The next case was 13.5 months after SBRT (Case 4). Sniped liver biopsy was performed for HCC; however, there was no cancer tissue, only non-cancer liver tissue. There was some focal fibrotic area (Figure 5). The majority was late-stage change, atrophy, and increased type I collagen fiber (Figure 5C). However, one part, “A” in Figure 5A, showed relative early-stage change degeneration of hepatocytes (Figure 5A,B) and falling out in silver impregnation and enlarged reticulum frame structure (Figure 5C).

Thus, the histological changes at an early stage for irradiation around HCC: at first, sinusoidal endothelium disappears and there is an irregular reticulum stain pattern of the sinusoid wall, followed by hepatocyte atrophy and fall-out with the enlargement of the reticulum frame structure. At a late stage, there is an increase in collagen type 1. 

## 4. Discussion 

To the best of our knowledge, this is the first report providing pathologic evidence for the phenomenon of FLR after radiotherapy. The results show that the pathological findings of FLR indicate an increase in hepatic artery in the portal vein area as complementary, due to the repair mechanisms in response to prolapse of hepatocyte and sinusoidal endothelium. Knowing the pathological and imaging changes associated with FLR in the liver surrounding the irradiated HCC is important to ensure safe and effective radiation therapy. 

According to previous reports [8,12,13], the mechanisms of FLR are considered as follows. Injured endothelial cells undergo apoptosis, although Kupffer cells are activated. Injured endothelial cells cause sinusoidal obstruction in irradiated liver. A hypoxic environment leads to hepatocyte death and inflammatory cell infiltration. Because Kupffer cells activate hepatic stellate cells, hepatic stellate cells lead to liver fibrosis [7,12]. Finally, hypervascularity in the arterial phase, namely FLR, is seen, because increases in hepatic arterial flow and decreases in portal blood flow occur due to sinusoidal obstruction and liver fibrosis [7]. In the pathological presentation of FLR, our findings are similar to those of previous reports [7,8,12], though some are different. In the current study, not only injured endothelial cells, but also the prolapse of hepatocytes, may be related to radiation damage. We believe that the degree of hepatic artery density in the portal area is important for understanding the increased hepatic artery as complementary, interpreted as reflecting FLR.

Pathological changes in FLR differ depending on the interval from radiation exposure [7]. In our study, acute, subacute, and chronic pathological changes were observed in five patients after radiotherapy. Thus, the intervals between radiotherapy and pathological investigation of FLR were obtained in various combinations. However, the current results were compatible with previously established results [5], stating that FLR began at a median of 3 months after radiotherapy, peaked at a median of 6 months, and disappeared 9 months thereafter. Background hepatic parenchymal enhancement around irradiated HCC persists for several months and may be observed as pseudoprogression when the contrast effect becomes nodular or enlarged [13]. Thus, we should pay attention to the misinterpretation of HCC recurrence.

The contrast enhanced hepatocyte image of EOB-MRI on FLR is similar to the functional damage to hepatocytes [14] and the alterations observed in pathologically atrophic hepatocytes.

Liver function according to the ALBI score significantly worsened after radiotherapy, indicating that the liver functional changes were affected by radiation damage. Although the relationship between FLR and liver function is not yet well known, Jung et al. [15] stated that patients with Child–Pugh B cirrhosis had a significantly greater susceptibility to the development of worse radiation-induced liver disease following SBRT. The status of the background liver is important, because two patients in our study with ALBI grade 2b showed a greater change in the ALBI score after radiotherapy (Table 1). 

The current study had a number of limitations. First, the number of patients in this study was small. A greater number of patients is required in further studies. Second, pathological proofs of FLR were obtained by needle biopsy (small specimen), except for one case of liver resection. Understanding the pathological mechanisms of FLR by needle biopsy may be difficult due to the small sample size. Third, some form of pathological hepatic change, except for SBRT, may have occurred given that two patients were modified by PTPE and TACE. Further analyses of untreated patients with HCC are desirable.

In conclusion, the pathological importance of FLR area is an increase in hepatic artery in the portal vein area due to the repair mechanisms in response to prolapse of hepatocyte and sinusoidal endothelium. In addition, liver function should be closely monitored since the ALBI score changes significantly following radiation therapy. Further investigations are recommended to more fully optimize hepatic radiation therapy. 

## Figures and Tables

**Figure 1 diagnostics-12-01072-f001:**
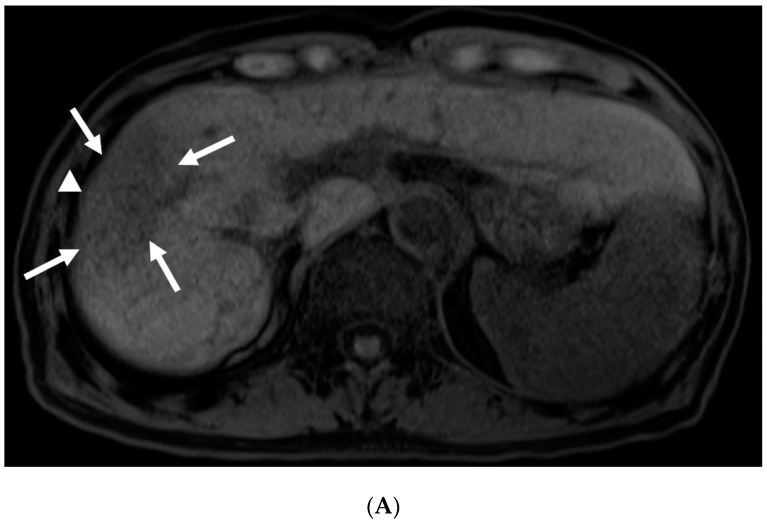
Gadolinium-ethoxybenzyl-diethylenetriamine pentaacetic acid enhanced magnetic resonance imaging (EOB-MRI) in a patient (Case 1) at 4.0 months after stereotactic body radiotherapy (SBRT). (**A**) Unenhanced T1 weighted MRI. Unenhanced T1 weighted MRI shows low signal for irradiated area. Focal liver reaction (FLR; arrows) and HCC (arrow head). (**B**) Arterial phase of EOB-MRI. Arterial phase of EOB-MRI shows hyper-enhancement as FLR (arrows) and HCC (arrow head), which means viability of HCC. (**C**) Hepatocyte phase of EOB-MRI. FLR (arrows) show low signal around HCC (arrow head), which is high signal in the hepatocyte phase of EOB-MRI.

**Figure 2 diagnostics-12-01072-f002:**
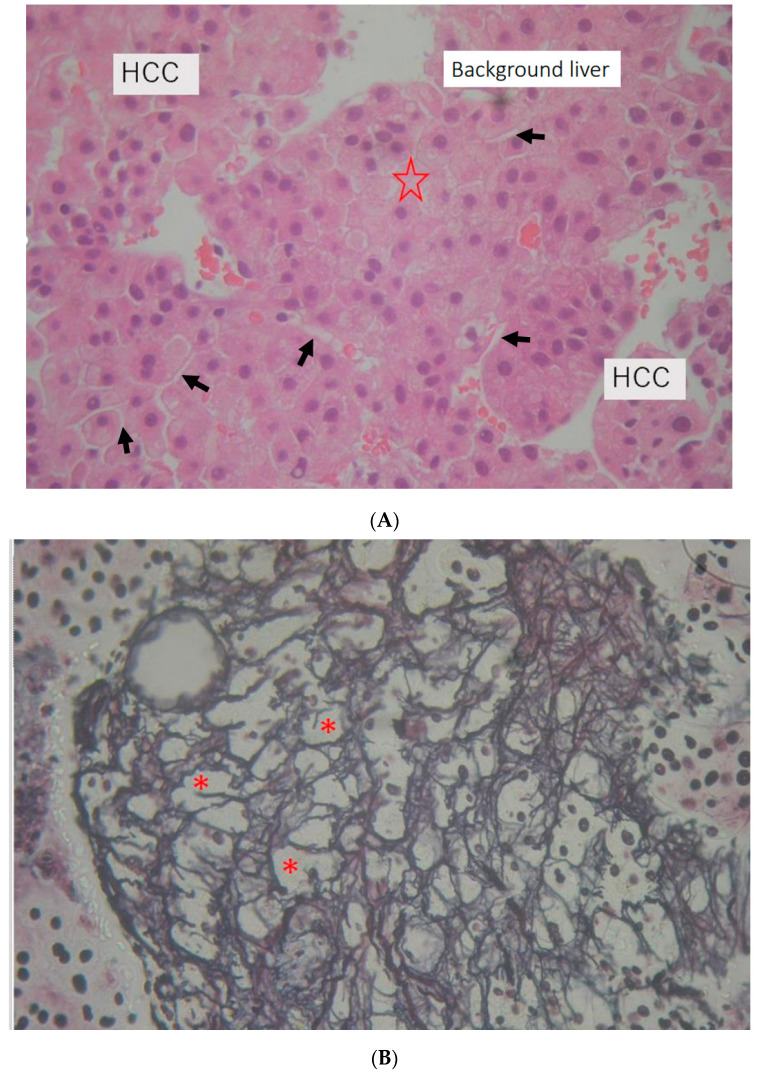
Pathology of liver biopsy (Case 1) at 1.5 months after stereotactic body radiotherapy (SBRT). (**A**) ×40 Hematoxylin and eosin staining; HCC and background liver HCC sniped biopsy include background liver tissue. Degenerated focus (☆). Endothelial cells (black arrows) are shown in the background liver. (**B**) ×40 silver impregnation stain; irregular reticulin frame work. Hepatocytes fell off (*) in the background liver tissue fragment.

**Figure 3 diagnostics-12-01072-f003:**
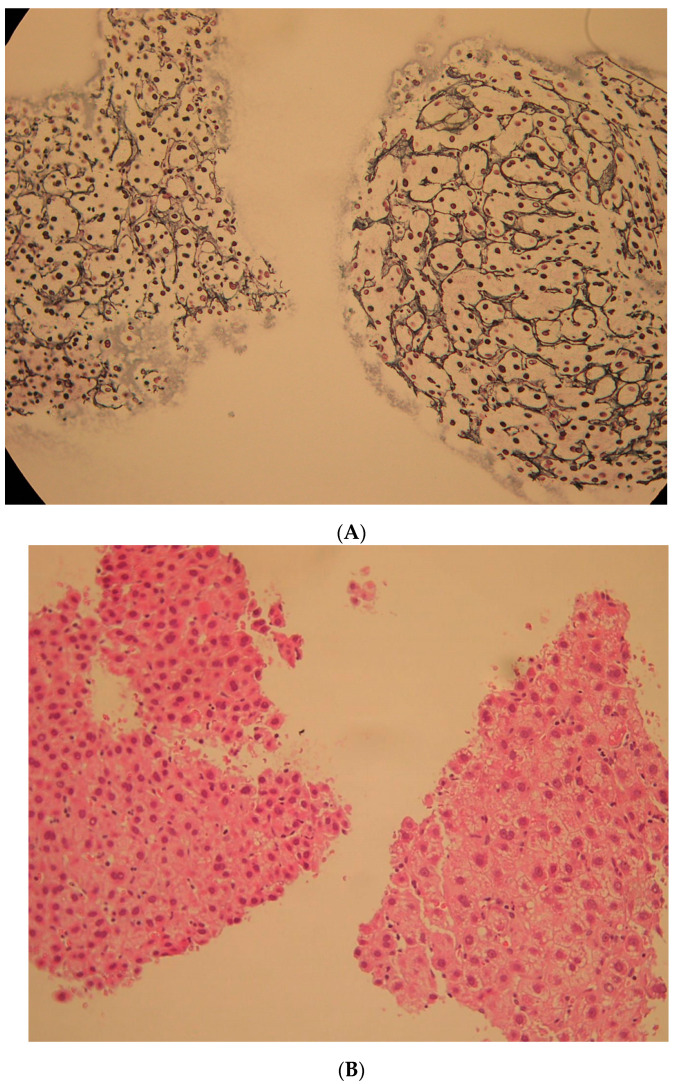
Pathology of liver biopsy (Case 2) at 3.0 months after stereotactic body radiotherapy (SBRT). (**A**) ×10 silver impregnation stain; HCC and background liver. From the liver tissue histology of the sinusoidal wall in the stain pattern, it was judged whether it was the liver that received radiation with the liver tissue histology of silver impregnation stain pattern of the sinusoidal wall. Background liver tissue fragment (right) and HCC fragment (left). (**B**) ×20 Hematoxylin and eosin staining; HCC sniped biopsy. Background liver tissue fragment (right) and HCC fragment (left). Both fragments received radiation. (**C**) ×40 hematoxylin and eosin staining; non-radiation liver tissue, as a control liver tissue. There are many endothelial cells (black arrows) in the sinusoids.

**Figure 4 diagnostics-12-01072-f004:**
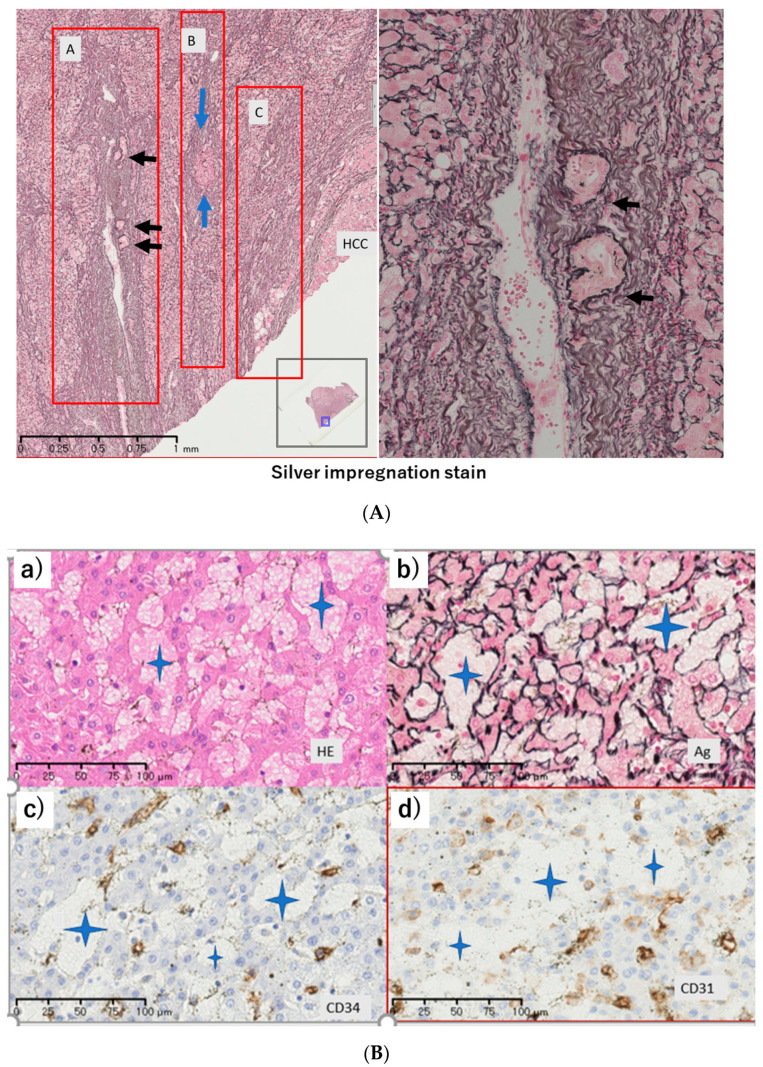
Pathology of resected liver (Case 3) at 4.0 months after stereotactic body radiotherapy (SBRT). (**A**) Silver impregnation stain. A, portal atrophy; B and C, portal vein disappearance and increased collagen fiber. Close to HCC nodule, hepatic tissue (hepatocyte and endothelium) fell off and so the portal tract (blue arrow) came close, and hepatic arteries (black arrows) subsequently increase. Background liver tissue fragment of A area (right; enlarged figure) showed hepatic arteries (black arrows) lined up inside the portal vein tract. (**B**) (**a**): Hematoxylin and eosin staining, (**b**): silver impregnation stain, (**c**): CD34, (**d**): CD31. (**C**) αSmooth Muscle Actin (SMA) immuno-histochemical staining. In portal tracts a little outside HCC nodule, portal vein disappeared and arteriole (positive at αSmooth Muscle Actin (SMA), arrows) increased. Close to HCC nodule, cluster of differentiation (CD)34, CD31 showed endothelium degeneration, and some disappeared. Hepatocyte was also lost, and then dilated space (Star mark) appeared.

**Figure 5 diagnostics-12-01072-f005:**
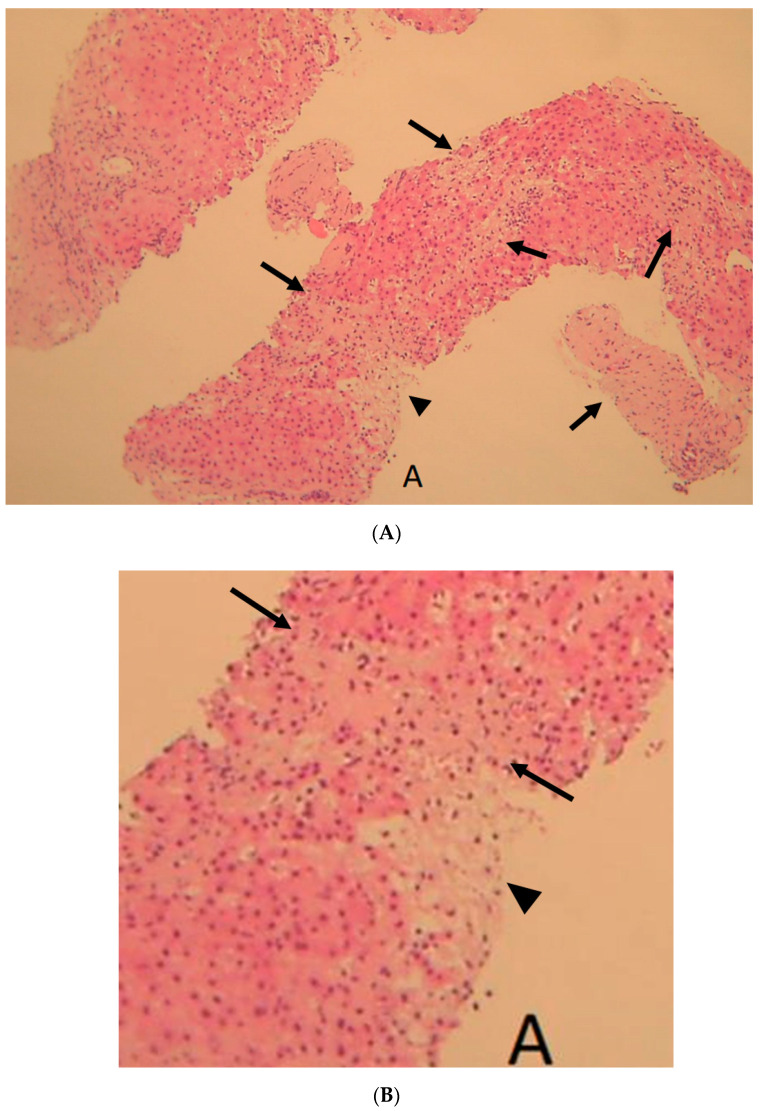
Pathology of liver biopsy (Case 4) at 13.5 months after hypofractionated radiotherapy (HFRT). (**A**) ×10 Hematoxylin and eosin staining; there are multiple fibrosis regions (arrows, as pale pink area) in liver biopsy fragment “A” area. (**B**) ×40 hematoxylin and eosin staining; irradiated background liver tissue fragment showed falling out of hepatocytes (arrowhead) in discolored part and fibrotic areas in pale pink part (arrows). (**C**) ×40 silver impregnation stain; in irradiated background liver tissue fragment “A” area of Figure 4A, hepatocytes fell out (※), and enlarged sinusoidal frame structure and increased collagen type 1 fibrosis (☆) were seen.

**Table 1 diagnostics-12-01072-t001:** Total bilirubin, Albumin and ALBI score: before and after radiation therapy.

No/Age/Sex	Hcc Seg/Size	RT Do/Fr/Da	Time	Pshology after RT		T-Bil/Alb (Pre-rad) ALBI Score/Grade	T-Bil/Alb (Post-rad) ALBI Score/Grade
1/81/M	S8/1.7 cm	SBRT	1.5 montha	poor	F1−2	0.98/4.2	0.4/3.6
		40Gy/5/5days				−2.79/1	−2.51/2a
2/55/M	S8/2.6 cm	SBRT	3.0 months	unknown	NASH	0.7/4.2	1.1/4.2−
		40Gy/5/5days				−2.86/1	−2.73/1
3/69/M	S8/2.3 cm	SBRT	4.0 months	well	F4	0.6/3.3	0.5/3.0
		54Gy/5/3days				2.14/2b	−1.93/2b
4/88/M	S3/3.2 cm	HFRT	13.5 months	poor	F2	0.5/3.2	0.8/2.9
		42Gy/14/18days				−2.10/2b	−1.72/2b
5/62/M	S2/1.3cm	HFRTs	14.0 month	well	F1	1.0/3.7	1.3/3.8
		42Gy/14/18days				−2.33/2a	−2.34/2a

Notes, M; Male, HCC; Hepatocellular carcinoma, Seg/Size; Segment/Size, RT, Radiotherapy. Do/Fr/Da; Dose/Fraction/Days. Time: Time interval between radiation therapy and pathology, T-Bil: Total bilirubin (mg/dL), Alb; Albumin (g/dL), ALBI; Albumin-bilirubin, pre-rad; pre-radiation therapy. post-rad; post-radiation therapy, SBRT, stereotactic body radiotherapy, HFRT, hypofractionated radiotherapy, poor; poorly differentiated HCC well; well differentiated HCC; Blood tests were obtained l2–26 days before radiation therapy and 18–40 days after radiation therapy.

## Data Availability

Not applicable.

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
