# Peer review of "Pathological Appearance of Focal Liver Reactions after Radiotherapy for Hepatocellular Carcinoma"

_diagnostics, 2022, doi:10.3390/diagnostics12051072_

Round 1

Reviewer 1 Report

Dear Editor,

                    In this manuscript authors reported their findings from a retrospective study assessing pathological reactions following radiotherapy against hepatocellular carcinoma (HCC) patients. The study is interesting and could benefit from the following:

  1. The abstract, particularly the conclusion should be improved.
  2. Table 1 – the table title should be improved to capture before and after radiotherapy.
  3. Figure 1 – the figure legend should appear below the figure. Also, descriptions for the subfigures in figure 1 should be incorporated in the figure legend instead of embedded in the subfigures.
  4. Comments 3 go for figures 3, 4 and 5.
  5. Authors should avoid or minimize use of “we, our, us etc”, e.g., Our study had a number, Our results show, we should pay attention, However, our results
  6. Some references are too old, consider replacement with recent ones, e.g., Herfarth KK, Hof H, Bahner ML, et al. Assessment of focal liver reaction by multiphasic CT after stereotactic single-dose radi- 335 otherapy of liver tumors. Int J Radiat Oncol Biol Phys. 2003;57: 444-51; Jung J, Yoon SM, Kim SY, et al. Radiation-induced liver disease after stereotactic body radiotherapy for small hepatocellular 362 carcinoma: clinical and dose-volumetric parameters. Radiat Oncol. 2013;8: 249.

Author Response

Please see attachment file (Word).

Reviewer 2 Report

I have carefully read the manuscript by Okada Franco et al. untitled: “Pathological appearance of focal liver reactions after radiotherapy for hepatocellular carcinoma” and here are my comments about it:

  • English language is good and the quality of manuscript also.
  • By reading it, I just could not envision the novelty of this work when compared to Sanuki-Fujimoto’s work.
  • Also, number of specimens is too slow in my opinion (n=5) to draw any significant conclusion.
  • May the authors increase the number of specimens?
  • In my opinion, authors did not enough emphasize on the novelty/plus presented in this work neither in the Introduction nor Discussion parts – This should be modified or at least more contrasted so that we (readers) could rapidly get the picture rapidly by reading it.
  • Also, Figures presentation should be modified to enhance clarity.

Author Response

Point-by-point responses to the reviewer 2 comments

Reviewer 2

Comments and Suggestions for Authors

I have carefully read the manuscript by Okada et al. untitled: “Pathological appearance of focal liver reactions after radiotherapy for hepatocellular carcinoma” and here are my comments about it:

  • English language is good and the quality of manuscript also.

→ Thank you for the kind reviewing the manuscript.

  • By reading it, I just could not envision the novelty of this work when compared to Sanuki-Fujimoto’s work.

→ As shown in the discussion of the manuscript, “In the pathological presentation of FLR, our findings are similar to those of previous reports,10-13 though some are different. In our study, not only injured endothelial cells, but also the prolapse of hepatocytes, may be related to radiation damage. We believe that the degree of hepatic artery density in the portal region is important for understanding the increased hepatic artery as complementary, interpreted as reflecting FLR.”

  • Also, number of specimens is too small in my opinion (n=5) to draw any significant conclusion.

→ This is the result of analysis of many specimens by experts in liver pathology, including special immunostaining, and we believe that this is an important study to analyze FLR after radiotherapy, although the number of cases is small.

  • May the authors increase the number of specimens?

→ The number of patients with recurrence after radiotherapy is very small, and the further pathological analysis of the FLR may be difficult, because it is necessary to perform RFA for the liver biopsy of FLR area. Unfortunately, further FLR biopsies would be difficult to perform.

  • In my opinion, authors did not enough emphasize on the novelty/plus presented in this work neither in the Introduction nor Discussion parts – This should be modified or at least more contrasted so that we (readers) could rapidly get the picture rapidly by reading it.

→ Thank you for the comment. As you mentioned, the additional References (new Ref, 7 and 8), are inserted for understanding increased arterial mechanism of FLR in Introduction.

 7             Takamatsu, S.; Kozaka, K.; Kobayashi, S.; Yoneda, N.; Yoshida, K.; Inoue, D.; Kitao, A.; Ogi, T.; Minami, T.; Kouda, W.; et al.Pathology and images of radiation-induced hepatitis: A review article. Jpn. J. Radiol. 2018, 36, 241–256.

8              Kim, J.; Jung, Y. Radiation-induced liver disease: Current understanding and future perspectives. Exp. Mol. Med. 2017, 49, 359.

And, the following sentence was inserted to the new manuscript (in the Discussion)

Additional sentence in the Discussion is as follows,

 “ To the best of our knowledge, this is the first report showing that provides pathologic evidence for the phenomenon of FLR after radiotherapy. “

  • Also, Figures presentation should be modified to enhance clarity.

→ Thank you for the comment. As you and Editor 1 mentioned, the figure legends of figures 1, 2, 3, 4 and 5 were optimized.

Round 2

Reviewer 2 Report

The authors replied well to my raised comments.